

# A history of the concept of time of concentration

Keith J Beven

[1]Lancaster Environment Centre, Lancaster University, Lancaster, UK

*Correspondence to*: Keith J. Beven (k.beven@lancaster.ac.uk)

**Abstract.** The concept of time of concentration in the analysis of catchment responses dates back over 150 years to the introduction of the Rational Method. Since then it has been used in a variety of ways in the formulation of both unit hydrograph and distributed catchment models. It is normally discussed in terms of the velocity of flow of a water particle from the furthest part of a catchment to the outlet. This is also the basis for the definition in the International Glossary of Hydrology. While conceptually simple, this definition is, however, wrong when applied to catchment responses where, in terms of how surface and subsurface flows produce hydrographs, it is more correct to discuss and teach the concept based on celerities and time to equilibrium. While this has been recognized since the 1960s, some recent papers and text remain confused over the definition and use of time of concentration. The paper sets out the history of its use and clarifies its relationship to time to equilibrium but suggests that both terms are not really useful in explaining hydrological responses. An appendix is included that quantifies the differences between the definitions of response times for subsurface and surface flows under simple assumptions that might be useful in teaching.

## 1 Introduction

The concept of the *time of concentration* of a catchment dates back to at least to Mulvaney (1851, reproduced in Loague, 2010) as the basis for estimating an appropriate time scale for rainfall duration in the Rational Method for estimating peak flows. More recently, time of concentration has been defined by the International Glossary of Hydrology (WMO, 1974; Johannsson, 1984) as the period of time required for storm runoff to flow from the most remote part of a drainage basin to the outlet. This definition has been used in a variety of hydrological texts (Richards, 1944; Chow, 1964; Haan et al., 1984; Maidment, 1993; Viessman and Lewis, 1995; Musy and Higy, 2004). Wikipedia gives a similar definition (https://en.wikipedia.org/wiki/Time_of_concentration), though elsewhere other computational definitions have been used (see, for example, the multiple definitions discussed in McCuen, 2009, and Grimaldi et al., 2012). This basic definition would, however, appear to be clear and unambiguous and, as such, has been widely cited in hydrological analysis and modelling.

There is, however, a problem with the concept of time of concentration in that the way it has been used in practice is generally in conflict with the Glossary definition. This confusion is apparent in the multiple definitions and methods of estimation for time of concentration that have been reported in the literature. Some recent reviews of estimation methods have been provided by McCuen (2009); Wong (2009), Almeida et al. (2014), Gericke and Smithers (2014); Grimaldi et al. (2012) and Michailidi





et al. (2018). Methods include the analysis of difference in different measures of timing for effective rainfalls and stormflows; empirical regression equations against catchment characteristics that can be used for ungauged catchments; and direct routing of runoff over the catchment topography and river network using estimated velocities. Grimaldi et al. (2012) even refer to the

time of concentration as a "paradox" and suggest that estimates by different methods can vary by 500%.

In fact, the Glossary definition reflects a fundamental misunderstanding of hydrological processes that has been part of the history of hydrology for more than 100 years.   The issue revolves around what is meant by the verb "to flow" in the standard Glossary definition.  The utility of the concept of time of concentration lies in its potential to provide an upper limit for the time scale of the hydrograph response of a catchment area.   However, it is not the flow velocities that should be used to define

the response times in a catchment, but the relevant surface, subsurface and channel flow celerities or wave velocities.  It seems that this might first have been suggested by Laurenson (1964) but has since been applied by many others (e.g. Morgali and Linsley, 1965; Eagleson, 1970; Beven, 1989, 2012; Wong, 2003; Saghafian et al., 2003; McDonnell and Beven, 2014).

Laurenson (1964, p.146) wrote: "*The "drop of water" concept of the runoff process described above is, however, unreal, as water on the ground and in the stream channels does not exist as a collection of drops, but as an amorphous mass.*

*Furthermore, were it possible to label individual molecules of water, it would be found that their paths and velocities of flow vary tremendously; some molecules would have an extremely short travel time, while others would never reach the outlet at all. We must therefore abandon the "drop of water" concept, and consider behaviour of water in the mass.*"

He did not, however, interpret the response of the amorphous mass explicitly in terms of celerities but goes on to suggest (p147): "*... it is in this sense that the term "travel time" is used in this paper. It is a storage delay time, and implies travel of*

*an effect rather than of a drop of water. The effect is transmitted by both wave movement and translation of the water.*"  He represents the effect by using nonlinear storage elements in the time-area discretisation of the catchment area.

It is the celerities or wave velocities that govern the hydrograph response to an input, for both surface and subsurface flows. Celerity is the speed with which a perturbation to a flow will propagate downstream (and in some cases upstream).   Celerity will generally be related to velocity but will depend on the flow conditions.   It will be different for different input intensities,

and different for rising limb relative to falling limb discharges.  In general it is necessary to make some assumptions about the nature of the flow in order to be able to estimate a local celerity, and the time it takes for a perturbation to reach the outlet of a hillslope or catchment (see Appendix for some surface and subsurface examples).

For the case of a dry initial condition and a steady input rainfall, the time from the start of rainfall to peak response is usually called the time to equilibrium and is obtained by integrating the celerity from upslope at time to t = 0 to a downslope outlet.

Eagleson (1970), for example, gives equations for time to equilibrium for surface runoff and refers to it as the time of concentration.   Beven (1982a,b,) does the same for subsurface flows and also refers to the time to equilibrium as time of



concentration. Wong (2003, 2009) treats the time of concentration as being equivalent to the time to equilibrium. In effect, this is how the time of concentration has often been used in practice when applied to the analysis and prediction of hydrographs, even if many explanations of the concept are still presented in terms of velocities rather than celerities. This is a result of the

historical development of the concept which the following text will explore in more detail.

**Early concepts of time of concentration**

Mulvaney (1851, p.23) notes the importance of "*the time which a flood requires to attain its maximum height during the continuance of a uniform rate of fall of rain*". He also notes that (p.24): "*This question of time, as regards any catchment, must depend chiefly on the extent, form and rate of inclination of its surface*". Mulvaney goes on to discuss the possibility of having a self-registering (recording) rainfall and stream gauges that would allow the development of relationships for estimating peak discharges in different circumstances. A little later, Kuichling (1889) was perhaps the first to define explicitly

a time of concentration as the time taken for water to flow from the furthest impermeable surfaces contributing to an urban drainage system.

In *The Elements of Hydrology* text of Adolf Frederick Meyer[1] (1917), in a section on Flood due to Rainfall, states: "*In general, the maximum flood due to rain will result from the greatest amount of most unfavorably distributed precipitation which may*

*be expected to occur over the entire tributary watershed within the time required for water from the remotest portion of the drainage basin to reach the point of observation. The time of concentration, in turn, depends upon the topography of the watershed and the size and slope of the water course*" (p. 309/310). Meyer provides no further explanation, which implies that this was already a term commonly understood by hydrologists and hydraulic engineers. Certainly, engineers designing drainage systems referred to the *point of concentration* at the end of a pipe system. Meyer (1917, p146), for example, notes

the requirement to estimate the lengths of time required for the runoff from a given precipitation to reach various points of concentration in a sewer system. Richards (1944, p33) defines this for a schematic catchment as the "*period of concentration*".

Given that time of concentration depends on the characteristics of flow pathways within a catchment, it is simple common sense to consider the distributed nature of those flow pathways in estimating catchment responses. It is also common sense

that the spatial patterns of rainfall and snowmelt inputs will have also affect the magnitude and timing of river discharges. It is not therefore surprising that these factors have been incorporated into hydrograph models for well over a century, even though the possibilities for doing so were limited by the availability of data and computational power in the days when computers were people.

---

[1] (1880 – 1962); see http://www.history-of-hydrology.net/mediawiki/index.php?title=Meyer,_Adolf_Frederick



The first such distributed model (that I know of) was proposed by Edouard Imbeaux[2] in 1892 for providing flood forecasts ("*Essai d'organisation d'un système d'annonce hydrometrique*") for the Durance River in south-east France following a series of large events in the Durance and Rhone in the period 1873-1890 (as well as major floods in 1843 and 1856). Imbeaux discretized the Durance catchment into zones of travel time to a point at which flood discharge predictions were required, and also elevation zones to allow for differences in rainfalls, the pattern of seasonal snowmelt, and consequent runoff generation.

By applying a form of degree-day snowmelt model and a simple local runoff coefficient in each of his discretized elements, the resulting storm runoff could be routed to the point of interest according to the specified time delays for each element. He expresses the concept in terms of routing water particles: "*notre molécule glissant depuis le pointe de sa chute jusqu'en A*" (where A is the point of interest). The result is what we would now call a time delay histogram or time-area histogram for routing runoff to the channel and then to the outlet.


Imbeaux recognized that the runoff coefficients would vary with both season and geology. He also proposes that the runoff coefficient would increase with mean rainfall intensity, and that for large events the soil will be largely already saturated. He also had information on the variation in rainfall totals for events over the Durance basin. In his analysis of travel times he looked at the propagation of the hydrograph peak down the river network for five exceptional events (10/1882, 10/1886,

11/1886, 10/1889, 3/1890) and found some consistency in the travel times, while noting that overbank flows would change the nature of the wave propagation. For surface runoff on the hillslopes he proposes a hydraulic representation for flow velocity as a square root function of flow depth (analogous to the Chezy or Darcy-Weisbach uniform flow relations), where flow depth is expressed as a specified fraction of rainfall depth in a time step. He also introduces a rule of superposition to allow for variations of rainfall intensity in different time steps and effectively introduces a time of concentration related to the

longest particle travel time ("*le temps que met la molécule la plus lente à faire son trajet*") as the number of hourly time step areas ("*courbes horaires*") required to represent the total response of the catchment.

Imbeaux realizes the approximate nature of his analysis, particularly in terms of assuming prior saturation of the soil so that there may be a delay in the response if that is not the case. He also notes that the end of the storm might be limited by the

infiltration of surface water into the soil, shortening the duration estimated from the time delay histogram. The importance of calculating effective storm runoff when there is rain on snow or rain with snow at higher elevations is also noted. He also regrets that the rainfall data from past events do not allow an analysis of the effects of rainfall variability over time, as he only has daily totals available.

---

[2] (1861-1943); see http://www.history-of-hydrology.net/mediawiki/index.php?title=Imbeaux,_Edouard



This rather remarkable study has all the elements of a modern distributed model, albeit simplified in terms of the resolution of the discretization and the nature of the runoff generation and routing methods employed. Imbeaux was not alone, however, in taking such an approach. Apparently independently, Ross (1921; reproduced in Loague, 2010) in Australia produced a similar time-area histogram approach to hydrograph prediction (see Figure 1). He notes that for constructed drainage schemes: "*All sewers and channels are designed for definite velocities, therefore starting from the discharge point for the whole system it is*

*quite easy to fix points on all pipes, channels, gutters etc., that are at the required time intervals from the discharge point*". For natural catchment areas, for "*rivers, rivulets and creeks* "*the velocities of flow can be found experimentally, and thus all points where time contours cross water courses can be accurately fixed*." He then suggests that for the hillslopes, lines can be drawn between these points by "*after examination of the country using the formula velocity* $= C\sqrt{rs} = k\sqrt{s}$ *where s is the average slope between the two time contours considered. Values of k would be required for the different classes of*

*country...*"(p.91). This is really rather similar to some forms of topographic analysis using digital elevation models carried out today. Similar forms of time delay histogram as distributed hydrological models were developed by Zoch (1934), Turner and Burdoin (1941) and Clark (1945) in the United States, and Richards (1944) in the UK.

      In each case, the time of concentration of a catchment was implicit in the number of time delay histogram elements used in

representing that catchment. In the simplest case, the velocities used to transform distances into time delays were assumed constant, for simplicity of computation, resulting in simple linear superposition of the delayed runoff from each element. Imbeaux (1892) did allow for velocity to vary nonlinearly with runoff magnitude on the hillslopes, but this can introduce difficulties under some circumstances if simple superposition is used (if runoff generation is higher in the upper part of a catchment it might be calculated as arriving at the outlet before runoff generation from the lower part of the catchment). Clark

(1945) did include a linear storage function into his time-area routing.

**Leroy K. Sherman and the Unitgraph**

      The time delay histogram approach of Imbeaux, Ross, Zoch, Clark and Richards has a number of inherent difficulties in

determining the areas in each time delay element, both in terms of defining the flow pathways to give the distances, and the effective flow velocities (or more correctly celerities) to transform distance into time, especially if that transformation might depend on flow rates in some nonlinear manner.

      Some of those difficulties were overcome by generalizing the time delay histogram to a catchment scale transfer function

derived from observed hydrographs. This generalization was first proposed by Leroy K Sherman[3] in 1932 as the unit-graph

---

[3] (1869-1954)





method. The transfer function relates a unit of effective rainfall as input to the same volume of storm runoff as output. "*The term effective rainfall means rain producing surface runoff*" (Sherman, 1932; also reproduced in Loague, 2010). He suggested that under some simplifying assumptions: "*By making use of a single observed hydrograph, one due to a storm lasting one day, it is possible to compute for the same watershed the runoff history corresponding to a rainfall of any*

*duration or degree of intensity*" (1932, p.54). Those assumptions include the stationarity of the unit-graph; linearity with respect to effective rainfall or storm runoff in excess of baseflow; and superposition of the contributions from successive inputs of storm runoff. Sherman does address the issue of how much of the rainfall should be considered as effective rainfall or storm runoff. He notes that "*percentages of runoff … appear to be very erratic*" (1932 p.57) but derives graphs of percentage runoff against storm rainfall for several catchments (see also Sherman, 1942). He also shows how to allow for

the effects of prior rainfalls, noting that the approach is "*rational, but the rule is empirical and only roughly approximate*" (1932 p.58).

The unit-graph approach is now more commonly known as the unit hydrograph method and is still widely used. Time of concentration has been considered as important to the unit hydrograph method in that it can be used to define the temporal

support for the unit hydrograph, often appearing as a parameter in defining some functional form, such as the triangular unit hydrograph. The triangle was first suggested as a simplification of the true time delay histogram by Zoch (1934) so that he could derive analytical solutions for hydrograph prediction from sequences of effective rainfalls when calculations had to be done by hand. The triangle was later suggested as a good approximation by the work of O'Kelly (1955) in Ireland; incorporated into the general linear theory of Dooge (1959)[4]; and later used widely, for example in the UK Flood Studies Report (FSR,

NERC, 1975) and Flood Estimation Handbook (FEH, Institute of Hydrology, 1999).

It has the advantage that only two parameters (three if an initial lag is needed) are required to define the shape of the triangle since the volume is constrained by definition to unity. Thus given a time to peak and a time of concentration the triangle is fully defined. This can be reduced to a single parameter if, as in the FSR and FEH, it is assumed that there is a fixed ratio

between them. Alternatively, given a time to peak and peak flow, the time of concentration, as the basal temporal support of the unit hydrograph can be calculated and does not need to be estimated separately.

It is worth noting at this point that, however, the unit hydrograph is derived, when integrated in time (known as the "S-curve" in unit hydrograph theory) provides a theoretical definition of the rising limb of the equilibrium hydrograph from a continuous

steady input. The unit hydrograph is in fact serving as a transfer function to transform the impact of a unit of effective rainfall over the hillslope into a hydrograph form. As recognized by Laurenson in 1964 (in the earlier quotation) and Morgali and

---

[4] (1922-2010); see http://www.history-of-hydrology.net/mediawiki/index.php?title=Dooge,_JCI_(Jim)





Linsley (1965) this is not the same as water droplets flowing to the outlet, even for the case of purely surface runoff. It is also not the same as a time to equilibrium that would be defined by the rising limb celerities under a continuous steady input, since a unit hydrograph derived from observations implicitly includes the nonlinear effects of falling limb surface and subsurface

celerities (e.g. Eagleson, 1970; Beven, 1982a,b), which can be expected to vary with input intensities (and antecedent conditions in the case of subsurface responses). To consider the unit hydrograph as a stationary linear transfer function for a catchment will, therefore, be an approximation, albeit that the stationarity assumption has often proven rather successful in real world applications (at least after mass balance is used to constrain the effective rainfalls in the analysis of hydrographs).

There is an enormous literature on methods of deriving unit hydrographs from observed rainfall and discharge data (including recent data-based transfer function methods that allow for some nonlinearity, e.g. Young, 2013) and fitting simple parametrically parsimonious forms, such as the triangle or gamma function. The advantage of using simple functions of this type is that the fitted parameters derived can be empirically related to catchment characteristics. The aim in doing so is to be able to estimate unit hydrographs for ungauged catchments, using parameters derived from gauged catchments. This has

included many studies that relate a time of concentration to catchment characteristics, starting with the regression approach of Kirpich (1940) for small catchments dominated by channel flow. The Kirpich equation has the form:

$$t_c = 0.0078(L^2/S)^{0.385}$$

where L is the length of the main channel and S is the slope. Kirpach did not derive the time of concentration from flow velocities, but from the translation of observed hydrographs. Thus this time of concentration is again effectively based on celerities and not velocities. Later the work of Nash (1959) in deriving such relationships was the basis for the approach taken in the FSR and FEH in the UK. There are many others. Reviews of such approaches, comparing estimates from different methods continue to the present day (e.g. Grimaldi et al., 2012; Gericke and Smithers, 2014; Michailidi et al., 2018) but often

without any clear discussion of velocities and celerities in the consideration of different methods and the subsequent estimates.

**Reconnecting to catchment topography - THE GIUH**

The unit hydrograph moved the analysis of the impulse-response of a catchment away from the spatial topographic

characteristics of the time delay histogram to a more generalized functional form. Starting in 1979, Ignacio Rodriguez-Iturbe and his colleagues started to reintroduce catchment geomorphology in the form of the geomorphological unit hydrograph (GUH; Rodriguez-Iturbe and Valdes, 1979; Rodriguez-Iturbe et al., 1979). The aim was ambitious: to provide an overarching





theory of the complex inter-relationships between runoff generation and the channel network, extending the seminal work of Horton[5] (1945); to explain the deep regularity of the channel network and catchment form (Rodriguez-Iturbe, 1993).


The GUH was based primarily on a statistical generalization of routing on the hillslopes and in the channel network. For the hillslopes routing was reduced to either assuming an instantaneous contribution of overland flow to the nearest channel or effectively a distribution function for the percentage of water drops added instantaneously over a catchment area to reach the outlet of a headwater channel (external network link) at time t. A one-parameter exponential distribution has been commonly

assumed. The rest of the channel network was handled in a similar way but taking account of the probability of reaches of a given order occurring in the channel network as represented by Horton's laws of basin composition. Using a Strahler ordering definition, ratios of stream numbers, lengths and areas for different stream orders can be assumed relatively constant. This provides a definition of the probability of water drops moving from one stream order to the next. A function for waiting times of droplets within each state can also be assumed, commonly again an exponential distribution for simplicity. The result is a

form of cascade of exponential distributions, but modified to allow for the order characteristics of the channel network.

Note that in this formulation, the nature of the response was still expressed in terms of the movement of conceptual water droplets. The integral of the instantaneous response function over time is then a time to equilibrium under a continuous steady input of effective rainfall. This is conceptualized, however, as a time of concentration allowing for the furthest water droplets

to arrive at the outlet (although under the exponential store assumptions the theoretical time to equilibrium is infinite such that definition of a finite time to equilibrium or concentration would therefore require truncation of the GUH at some point).

The times scale of the GUH also requires some additional assumption. Horton's laws relate reach numbers, lengths and areas to stream order, but do not give a relationship that allows a time scaling. In one sense this does not matter in that the parameters

of the distribution functions of the theory can be fitted to data as time constants. However, taking advantage of work by Leopold and Maddock (1953), that showed that mean stream velocities increase only slowly with catchment area and stream order, it is possible to relate length characteristics to time under the simple assumption of a constant mean stream velocity. Rodriguez-Iturbe and Valdes (1979), by assuming that the GUH could be expressed as a triangle, carried out a regression analysis over a wide range of theoretical responses to identify equations for peak flow and time to peak of the response function

that involve the Horton ratios, stream order lengths and the mean stream velocity. The mass balance constraint under the triangular assumption means that time of concentration does not need to be considered explicitly, it is defined by the peak flow and time to peak. Rosso (1984) fitted a 2 parameter gamma distribution (equivalent to the Nash cascade of linear stores) to the GUH transfer function where the time constant parameter also involves the mean stream velocity, but which can also be

---

[5] (1875-1945); see http://www.history-of-hydrology.net/mediawiki/index.php?title=Horton,_Robert_Elmer



fitted directly to observed effective rainfall and direct runoff data.  Note that this is still being expressed in terms of velocities
rather than celerities.

Rodriguez-Iturbe et al. (1979) noted that because of this velocity parameter, the GUH might be expected to vary from event
to event and also within an event.   They suggested, based on model results, that for any given event, the GUH could be
characterized by the velocity at the peak flow.   Later work served to replace the velocity parameter with its dependence on
the intensity and duration of the rainfall excess in what was called the geomorphoclimatic theory of the instantaneous unit
hydrograph (Rodriguez-Iturbe et al., 1982).   In the derivation, kinematic wave theory is used to predict the peak flow velocity
in a given channel order given storm intensity and duration, under the assumption that the effective rainfall is of sufficient
duration to exceed the time to equilibrium of a 1$^{st}$ order stream (this was later relaxed by Nowicka and Soczynska, 1989, who
extended the analysis to partial-equilibrium hydrographs). Storm intensities and durations will vary from storm to storm
however and under assumptions about their distribution the stochastic distribution of the GUH peaks and time to peaks can be
obtained by derived distribution theory (e.g. Rodriguez-Iturbe, 1993).   This can be taken further to determine the flood
frequency characteristics, given some functional way of estimating effective rainfall for an event from rainfall statistics (e.g.
the infiltration capacity method used in the derived distribution approach of Eagleson, 1970).

There are three features to note, in general, about the GUH approach.   The first is that it deals primarily with the channel
routing component of catchment response.  It does not explicitly deal with how much of the rainfall becomes effective rainfall,
except in the infiltration excess approach to estimate flood frequencies by derived distributions noted above (Beven, 1986).
The second is that although there is a time of concentration implied by the triangular approximation to the impulse-response
used in the regressions against basin order ratios, it is never explicitly considered because of the mass balance constraint.   This
is perhaps just as well given that the full expression of the GUH has an infinite tail.

Thirdly, there is the treatment of the velocity parameter that provides the time scaling of the impulse response.   Throughout
the GUH literature this is expressed as a mean channel flow velocity.   When the velocity is allowed to vary with storm
characteristics, the mean channel flow velocity at the peak has generally been used.  Where the kinematic wave approximation
to the velocity/discharge relationship was invoked, at the 1$^{st}$-order basin scale the peak mean channel flow velocity at the time
of equilibrium was determined.   This is somewhat ironic, because estimating that time to equilibrium in a headwater involves
a kinematic wave celerity (as given by Henderson and Wooding, 1964, Morgali and Linsley, 1965; and Eagleson, 1970) but
the use of the maximum velocity still seems to invoke thinking that is firmly rooted in the velocities and travel times of water
droplets in overland flow rather than wave velocities or celerities. It also takes no account of falling limb celerities.  GUH
theory therefore consistently muddles these different concepts.  The historical context is perhaps important here.  The GUH is
still based on thinking about stormflow as a surface runoff but in the same year that the first GUH papers appeared, Sklash and
Farvolden (1979) published their environmental isotope tracer paper that showed that hydrographs could be dominated by



stored water and that overland flow could be pre-event stored water displaced by the storm rainfall inputs. This paper had a greater effect on perceptual models of catchment response than some of the earlier tracer and geochemical-based work on
stored water contributions to the hydrograph.

**The coming of digital elevation data: distributed time of concentration calculations**

One of the reasons for the generalisations of the UH and GUH approaches to modelling catchment responses was that before the 1980s there was not general access to digital terrain data or digital elevation models (the topographic analysis that underlay the topographic index used in Topmodel, for example, was originally carried out manually, see Beven, 2012). Once such data did become available it also became possible to consider catchment geomorphology more directly in calculating times of concentration on a more distributed basis. The most common approach was to use gridded topographic data, with a calculation
of incremental travel times in each grid and channel reach. It seems that in doing so there was again a general confusion as to what velocities should be used in such a calculation; while in practical applications the velocity was again used as a calibration parameter.

Zuazo et al. (2014) provide a general review of these approaches, together with a comparison of methods applied to some
hypothetical cases. For their study, they define the time of concentration as "*the time at which the catchment or plane is in equilibrium or equivalently the travel time to the downstream end of the plane (x=L) of a wave originating at the upstream end of the plane*" (p.1318), noting that solutions for this time to equilibrium based on wave velocity or celerity had been previously given by Morgali and Linsley (1964), Eagleson (1970) and Singh (1976). They conclude that it is important to take account of upslope inputs to have a more robust estimate of the resulting unit hydrograph to avoid additional sensitivity
to grid size and input rates. Aron et al (1991) apply kinematic wave celerities to determine what they call the time of concentration on a fractal topography from hillslopes to rills to a channel network. Saghfaian et al. (2002) also base their distributed time delay histogram on estimated celerities for surface runoff and provide one of very few discussions of the difference between using velocities and celerities (see also Wang, 2003, for the single flow plane case). They note that their calculations can take account of spatial variability in effective rainfalls and result in time-area histograms that are non-
stationary with flow conditions.

In contrast, other recent studies continue to approach the time of concentration from the point of view of the International Glossary definition based on velocities (e.g. Du et al., 2009). Manoj and Xing (2014) and Li et al. (2018) use particle tracking techniques based on calculated distributed velocity fields to determine the travel times of water particles, with time of
concentration estimated from the longest travel times. The use of velocities in such distributed approaches is specifically criticized by Saghafrian and Noroozpour (2010).



**Velocities, celerities and the time of concentration**


This discussion of the historical development of the time of concentration concept has demonstrated that the considerable confusion over the use of the term still persists. As noted earlier I have been part of that problem. In the papers of Beven (1982a,b) time to equilibrium is correctly used in assessing the hydrograph responses but, following Eagleson, 1970, is referred to as time to concentration, thus conflicting with the WMO Glossary definition. In contrast, other studies have evaluated time

of concentration from flow velocities, but have then used this as if it was a time to equilibrium in predicting hydrograph responses. Returning to Imbeaux (1898), he effectively used velocity estimates in his subcatchments and celerities in the channels.

If we are interested in hydrograph responses, it should be clear that we should be not so much concerned with the time of travel

of an input water particle from the farthest reaches of the catchment to the outlet as with the time it takes for the effect of that input to have an effect on the output. This is a function of celerity rather than velocity, where celerity is the speed at which pressure waves can move through the system (or the forward and backward characteristics in the case of the St. Venant equations for channel flow, see Appendix).

Thus, for clarity, It is necessary to replace reference to velocities by reference to celerities or wave velocities. The nonlinear changes in celerity over the course of a hydrograph are partly what lead to the asymmetry of hydrograph shape. Thus any interpretation of the unit hydrograph in terms of travel times of water particles is neglecting the changing nature of the celerities over the rising and falling limbs of the hydrograph. Note, however, that the dependence of celerity on flow magnitude also suggests that the linearity assumptions of the unit hydrograph and its time of concentration support will not be generally valid,

as previously recognized in the geomorphoclimatic UH theory. The most widely cited example of this is the demonstration of the change of unit hydrograph with peak discharges in a small catchment by Minshall (1960). However, as Imbeaux showed in his analysis of flood peaks in the Durance, the assumption of a constant celerity might at least sometimes be a useful first-order approximation. This was also shown for the case of the channel tracing experiments of Pilgrim (1977) and for upland channel reaches by Beven (1979), for one particular velocity-discharge function, albeit that velocities change nonlinearly with

discharge (see Appendix, equation [A18]).

**Mathematical celerities and natural hillslopes**





The quantitative analysis of the difference between velocity and celerity responses presented in the Appendix is intentionally simple. It is intended to make it quite clear for pedagogical purposes why the traditional definition of time of concentration is inconsistent with the way in which it is commonly used in practice. It has been used in the past to relate estimates of time to equilibrium to hillslope form expressed in terms of convergence and convexity (e.g. Morgali and Linsley, 1965; Singh, 1976; Sabzevari et al. 2013). It is, however, a mathematical result subject to the various sets of assumptions that underly the

analytical solutions presented. It therefore begs the question of whether the mathematistry might be applicable to natural hillslopes. The approximations might not be adequate in hillslopes with 3D patterns of heterogeneous soil and vegetation characteristics, surface roughness and microtopography, preferential flowlines, soil moisture deficits, depths to bedrock and losses through some (sometimes vaguely) defined lower boundary condition. These heterogeneities do suggest that there will be a complex time and space variability in both velocities and celerities, with the potential for different values and different

degrees of diffusion in different flow pathways depending on local structures and flow rates.

   Given that we now know that in many catchments the water making up the storm hydrograph is water stored in the catchment prior to an event, it would seem that the standard WMO Glossary definition has little relevance to hydrograph analysis. This understanding implies that there is no simple delay mechanism for water flowing towards the outlet, but a complex interaction

between event water and stored water. Even for overland flow processes, the distance over which such interactions occur might be rather local (e.g. the suggestion that microtopography can play a role in old water displacement in Iorgulescu et al., 2007 and the estimate of a mean flow path length for surface runoff of 1m in the work of Bergkamp, 1998). This means that water falling at the furthest distance from the outlet might not be expected to contribute to the storm hydrograph for that event, even if some overland flow is generated at that point. In fact, such tracer studies suggest that the actual times for water to

flow from the furthest point in a catchment to the outlet may, at least in humid catchments, be many years. Indeed, Berghuis and Allen (2019) have recently suggested that while the storm hydrograph might predominantly be made up of pre-event water, it might still be (on average) younger than the mean residence time of stored water in the catchment (see also Gallert et al., 2019). That stored water will include water that fell far from the outlet in past events.

What therefore can we take from this analysis that might be applicable to natural hillslopes? The basic concept that the response of the hillslope as seen in the hydrograph will be faster than water can flow from the furthest point upslope will hold even in the case of a steady input. Thus, time to equilibrium could be expected to be shorter than the time of concentration in the Glossary definition (see Appendix). This will hold for both surface and subsurface flows and for all representations of flow processes for which velocity increases with depth of flow or saturation. The difference between celerities and velocities

is likely to be very large in soil with small effective storage deficit above the water table (i.e. in near saturated conditions, or with a significant capillary fringe).

Similar issues can apply in the unsaturated zone, where the local celerities associated with film flow (including in preferential flow pathways) can also be faster than the local velocities, resulting in the potential for the displacement of stored water

(Bogner and Germann, 2019), though in that case there is the issue of whether the local wetting fronts reach the water table before being overtaken by a following drying front, especially where the saturated zone may be deeper in the upper part of a hillslope.  In structured soils it is also possible that both mean pore water velocities and celerities will be affected by bypassing in larger voids, such that the effective storage deficit might include infiltration into soil peds.

Thus while it may be difficult to provide a complete description for the flow processes on complex hillslopes it is suggested that the conceptual consequences of the differences between celerity and velocity responses are important and should be incorporated into hydrological teaching in future.   It is also worth noting that since celerities are generally faster than mean areal flow velocities this is also a (partial) explanation for the displacement of stored water in forming the hydrograph (Beven, 1989; McDonnell and Beven, 2014).


However, the relevance of the time to equilibrium concept can also be questioned, particularly when rainfall durations are shorter or intensities are nonuniform, because of the expectation that celerities will be different on the rising limb and falling limbs of a hydrograph.  Certainly, now that we are much less constrained by computational limitations, it will be better to predict the changing celerities during an arbitrary event directly than determine a unit hydrograph from only the S-curve of the

rising limb to equilibrium.

**Teaching the concept of time of concentration in future.**

In the same way that hydrologists should really avoid using the rather desperate technique of hydrography separation (see the section on choosing a method of hydrograph separation in Beven, 1991), it might be better to avoid the use of time of concentration in the IGH glossary definition and time to equilibrium concepts completely.   The idea of a maximum time of travel for a water droplet is an attractive one, but it does not survive critical analysis in terms of predicting hydrographs.  It is clear, however, that the use of catchment characteristics to predict a "time of concentration" as a basal temporal support for

the unit hydrograph continues to this day as one approach to predicting the response of ungauged catchments.

In one sense this is acceptable in the context of the identification of a suitable transfer function from observations since this will implicitly take account of the relevant celerities in the catchment without invoking any assumptions about water velocities (water velocities only become important if there is an additional requirement to estimate residence and transit times that will

generally be much longer than the hydrograph time scale).  Where we can go wrong, however, is in keeping any association





with the Glossary definition of time of concentration in terms of water travel times, and in the GUH time scale parameter as a velocity rather than as a celerity.

If it is necessary to discuss the concept of time of concentration therefore, physical correctness requires that the Glossary definition be abandoned and replaced by a discussion of celerities and time to equilibrium (while noting that falling limb celerities will be different from those that define a time to equilibrium under a steady uniform input).   The kinematic wave theory provides a simple framework for showing how celerities can be greater than velocities, and how they might vary for different intensities and durations of rainfall and antecedent conditions.   This has already been included in some texts, starting with Eagleson (1970).   Checking other hydrological text books, Overton and Meadows (1976) and Bedient and Huber (1988)

specifically state that time to equilibrium should be used in estimating catchment responses, while Brutsaert (2005) mentions only time to equilibrium and not time of concentration. Hornberger et al. (1998) do not discuss either. Haan et al. (1984), Viessman and Lewis, (1995) and Musy and Higy (2004), in contrast, still refer to time of concentration in the glossary definition.  This has also continued in the recent papers of, for example, Li et al. (2018) and Michailidi et al. (2018).   More embarrassingly, Shaw et al. (2011) following earlier editions, treat time of concentration as the longest travel time based on

velocities on both a catchment and in a pipe network while more correctly discussing celerities elsewhere.

In teaching, however, it should also be pointed out that the kinematic analysis presented is rather over-simplified, in that on real hillslopes the flow relationships will be more diffusive (kinematic shocks are rare in real flows) and will be spatially variable depending on soil, topography and input sequences.   The hydrograph will then reflect the integral effects of the patterns of inputs and celerities in space and time.   What needs to be emphasised, however,  is that time of concentration in

the Glossary definition is irrelevant to catchment hydrograph responses.   The frequent reference to time to equilibrium as a time of concentration has only confused the issue in both teaching and practice (and, of course, a steady uniform input is really rather rare in reality anyway).  If we can explain the hydrograph responses arising from surface and subsurface flows in terms of time-variable celerities, including the variation with patterns of inputs and antecedent conditions, perhaps both of these

terms should be abandoned forthwith in hydrological teaching for the 21$^{st}$ Century.

## Acknowledgements

KB would like to thank Prof. Charles Obled for bringing the work of Edouard Imbeaux to his notice and providing pdfs of the relevant pages of the 1892 paper.    This paper was instigated by considering the problem of changing catchment responses

under natural flood management projects in the Q-NFM project led by Dr. Nick Chappell of Lancaster University (NERC grant no. NE/R004722/1).





## Appendix: Time of Concentration and Time to Equilibrium for flow on a plane or channel reach.

To illustrate the difference between time of concentration and time to equilibrium in the simplest way, let us assume that the
downslope subsurface and surface flows can be represented by a kinematic approximation defined by simple power law
functions between mean flow velocity at a point, v, and depth of flow, h (we ignore the unsaturated zone, but see Beven,
1982a,b, for a kinematic treatment of wetting and drying front propagation):

$$v = ah^b \qquad\qquad [A1]$$

where a and b are parameters. Beven (1982a) gives example parameters for downslope saturated subsurface flows for both
power law and exponential profiles derived from field data. The kinematic wave equation is then defined by combining this
function with the continuity equation:

$$\varepsilon \frac{\partial h}{\partial t} = \frac{\partial q}{\partial x} + i \qquad\qquad [A2]$$

where $q = \int_0^h v\,dh$ is the local specific discharge, $i$ is a local input rate and $\varepsilon$ is an effective storage coefficient for the change

in depth of flow with change in storage volume. Since, with these definitions, $\frac{dq}{dh} = v$, so that:

$$\varepsilon \frac{\partial h}{\partial t} = ah^b \frac{\partial h}{\partial x} + i$$

or

$$\frac{\partial h}{\partial t} = c \frac{\partial h}{\partial x} + \frac{i}{\varepsilon}$$

where c is the celerity (with dimensions LT$^{-1}$). Thus for the power law flow relationship when

$$c = \frac{ah^b}{\varepsilon} \qquad\qquad [A3]$$



For mean Darcian flow in subsurface flow on a slope $\beta$, with distance measured along the slope, and hydraulic conductivity as
a power function of depth of saturation:

$$a = K_o \sin \beta$$

where $K_o$ is the hydraulic conductivity when $h = 1$, $b$ defines how quickly hydraulic conductivity increases with depth of
saturation, and $\varepsilon$ is the storage deficit above the water table per unit depth of rise or fall of $h$.

Then, for the subsurface flow case (assuming that $\varepsilon$ is approximately constant):

$$c = \frac{K_o \sin \beta h^b}{\varepsilon} \qquad [A4]$$


For these relationships, at any $h$ the ratio of the celerity and local mean water particle velocity, $v_p$, can be determined where

$$v_p = \frac{K_o \sin \beta h^b}{n} \qquad [A5]$$

Where $n$ is the porosity of the soil (again assumed constant for simplicity).   The local ratio for celerity to mean pore water
velocity is therefore $n/\epsilon$, and since the effective storage $\varepsilon$ is always less (and sometimes much less in a wet soil) than the
porosity $n$, celerity is always faster than the mean pore water velocity which itself is greater than the mean Darcian velocity,
$v$.   Integrating to the length of a flow plane gives the time to equilibrium (dependent on $c$) and the time of concentration (here
as the integral of mean pore water velocity for drops of water ($v_p$), assuming $h = 0$ when the recharge $i$ reaches the base of the
soil profile at $t = 0$ (note that Beven, 1982a,b, gives a more complete solution that allows for travel time of a wetting front
through the unsaturated zone and filling to saturation for depths when $K_s < i$).

Equations [A1] to [A5] set out the basis for evaluating celerities and velocities in subsurface flow on a hillslope plane using
kinematic wave theory.   Under the simplifying assumptions of a power law and constant values of effective storage and soil
porosity the local ratio of celerity to mean pore water velocity is then given by:

$$\frac{c}{v_p} = \left\{ \frac{K_o \sin \beta h^b}{\varepsilon} \right\} / \left\{ \frac{K_o \sin \beta h^b}{n} \right\} = \frac{n}{\varepsilon} \qquad [A6]$$





where $n$ is the porosity of the soil. It was noted that since the effective storage $\varepsilon$ is always less (and sometimes much less in a wet soil) than the porosity $n$, celerity is always faster than the mean pore water velocity which itself is greater than the mean Darcian velocity, $v$. Integrating along the length of the plane we can derive an analytical solution for both the time of equilibrium (

$$TOC_{vp} = n \left[ \frac{1+b}{K_o \sin \beta} \right]^{\left(\frac{1}{1+b}\right)} i^{\left(\frac{-b}{1+b}\right)} L^{\left(\frac{1}{1+b}\right)} \qquad \text{[A7]}$$

$$TTE_c = \varepsilon \left[ \frac{(1+b)}{K_o \sin \beta} \right]^{\left(\frac{1}{1+b}\right)} i^{\left(\frac{-b}{1+b}\right)} L^{\left(\frac{1}{1+b}\right)} \qquad \text{[A8]}$$

Thus the time of concentration and time to equilibrium are also in a fixed ratio of $n/\varepsilon$ in this case. The wetter the soil, and therefore the smaller the effective storage $\varepsilon$ relative to the porosity $n$ the greater will be the difference between the two response times. Some results for constant values of $\varepsilon$ and different input rates are shown in Figure A1. It would not be expected that $\varepsilon$ will stay constant during an event, but would be initially low in a wet soil due to any capillary fringe or macroporosity, becoming larger depending on the form of the wetting front during recharge. TOC will, however, necessarily be larger than TTE.



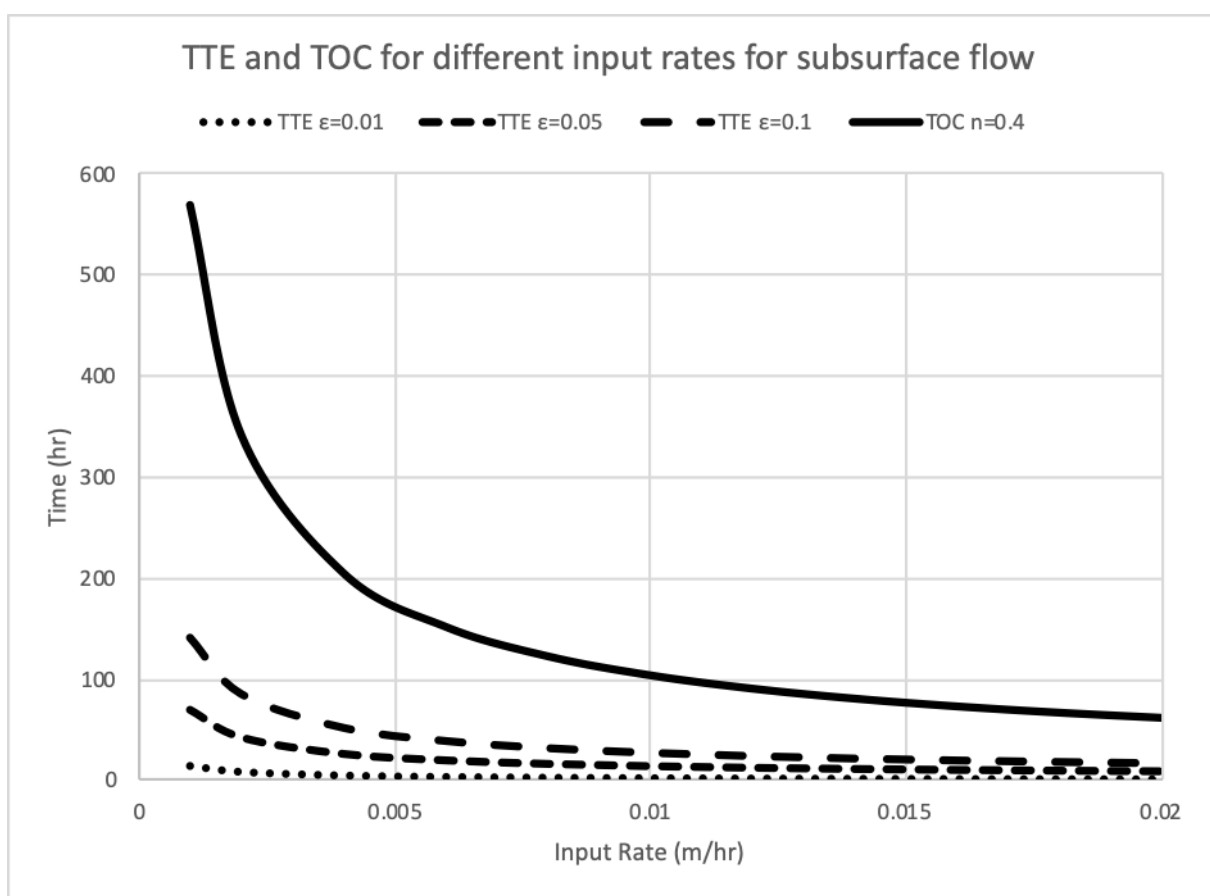

**Figure A1. Time of Concentration (TOC solid line) and Time to Equilibrium (TTE dashed lines) for subsurface flow on a hillslope of 100m for different input rates and values of effective porosity (TOC is constant for each input rate) ($K_o$= 2 m/hr, sin $\beta$=0.05)(Equations [A7] and [A8]).**


.

For the special case of a uniform profile of hydraulic conductivity (b=0), time of concentration and time to equilibrium are no longer dependent on the input rate $i$ (mean pore water velocities and celerities are constant regardless of depth of saturation).

Beven (1982a,b) gives more detail for this case in non-dimensional coordinates.

$$TOC_{vp} = n\left[\frac{1}{K_o \sin \beta}\right] L \qquad \text{[A9]}$$



$$TTE_c = \varepsilon \left[ \frac{1}{K_o \sin \beta} \right] L \tag{A10}$$


For the case of a confined saturated layer with $\varepsilon \ll n$ (as in a saturated pipe) then the celerity will approach infinity.

For surface flow that is treated as a uniform sheet flow that conforms to a Darcy-Weisbach type relationship then mean velocity at any depth of flow is given by


$$\bar{v} = C(\sin \beta)^{0.5} h^{0.5} \tag{A11}$$

So that

$$q = \bar{v}h = C(\sin \beta)^{0.5} h^{1.5} \tag{A12}$$


where C is a roughness factor. Note the difference in the definition of $v$ from the subsurface case. Following normal practice, the velocity relationship takes account of the effective porosity for surface flow implicitly in the expression for velocity and water droplets are assumed to travel with an average velocity of $\bar{v}$. These types of kinematic relationship can also be used to derive hillslope and channel times of concentrations analytically, given values of the parameters (e.g. Henderson and Wooding,

1964; Eagleson, 1970; Beven, 1982; Wong and Chen, 1997). In this case celerity as $\frac{dq}{dh}$ is given by

$$c = 1.5 \, C(\sin \beta)^{0.5} h^{0.5} \tag{A13}$$

Thus


$$\frac{c}{v} = \{1.5 \, C(\sin \beta)^{0.5} h^{0.5}\} / \{C(\sin \beta)^{0.5} h^{0.5}\} = 1.5 \tag{A14}$$

so that $c$ will always be greater than $\bar{v}$. Integrating along the plane again gives expressions for the time of equilibrium (dependent on $c$) and the time of concentration (here as the integral of mean velocity $\bar{v}$ for drops of water).


$$TOC_v = \frac{3}{2} \left[ \frac{1}{C(\sin \beta)^{0.5}} \right]^{2/3} i^{-1/3} L^{2/3} \tag{A15}$$


$$TTE_c = \left[\frac{1}{C(\sin \beta)^{0.5}}\right]^{2/3} i^{-1/3} L^{2/3} \qquad [A16]$$

Taking the ratio of the two times gives a ratio of 1.5 longer for the time of concentration. Some indicative times are shown in Figure A2.

Thus in both surface and subsurface flow an input of water can have an impact on the hydrograph more rapidly than the water "droplets" can flow from the farthest distance to the output. Both, however, vary with the intensity of the input. Time of concentrations for surface runoff will vary with the way in which friction losses are expressed, including any allowance for effective porosity when flow is through a vegetation cover. For subsurface flow it will also vary with the hydraulic conductivity profile in the soil (see for example, Beven, 1982a,b). Note that in this case we consider only recharge to the water table at the base of a soil profile that is deep enough that the soil does not saturate to the surface. Henderson and Wooding (1964) also considered a diffusive wave solution for steady state subsurface flow, with different downstream boundary conditions, but did not derive expressions for the time of concentration or equilibrium.

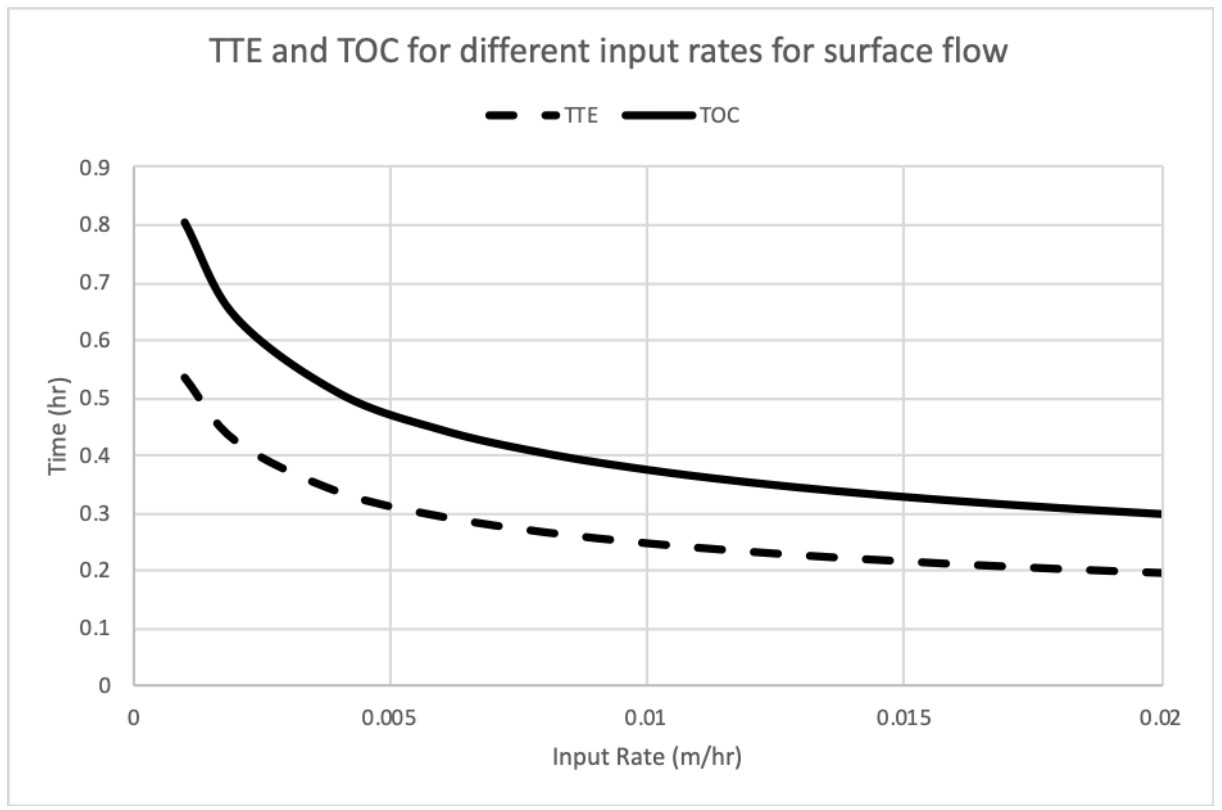



**Figure A2. Time of Concentration (TOC solid line) and Time to Equilibrium (TTE dashed line) for surface flow on a hillslope of 100m for different values of input rate (C=10 $m^{0.5}$/s, sin $\beta$=0.05) (Equations [A15] and [A16]).**


For channel flows, the 1D St. Venant equations are hyperbolic partial differential equations that have both upstream and downstream characteristics in sub-critical flow conditions. In this case the downstream celerity is

$$c = v + \sqrt{gh} \qquad [A17]$$


where $g$ is the acceleration due to gravity. Thus, $c$ will again always be greater than the mean cross-sectional flow velocity, but this will depend on the cross-sectional average depth. The deeper the flow, the greater the ratio $c/v$ will be. The difference will be small for shallow surface sheet flows, but the velocity will also then be small so the difference might still be significant.

For small rough headwater channels for which the kinematic wave equation might still be a useful approximation, Beven (1979) also showed that the change of velocity with discharge $Q$ could be described by

$$v = \frac{Q}{A} = Q\left\{\frac{a}{Q+k}\right\} \qquad [A18]$$

where $a$ and $k$ are parameters and $A$ is the local cross-sectional area of the channel. Support for this type of relationship in small channels is also given by Figure 2 of Pilgrim[6], 1977). For this specific function the celerity is

$$c = \frac{dQ}{dA} = a \qquad [A19]$$

This is one example of where the time of equilibrium for a headwater channel reach of length $L$ could be considered constant
for all discharges while reflecting a nonlinear velocity discharge relationship. Thus there is still a difference between the time to equilibrium and time of concentration as:

$$TTE_c = \frac{L}{a} \qquad [A20]$$

---

[6] (1931-2015); see http://www.history-of-hydrology.net/mediawiki/index.php?title=Pilgrim,_David_H



The time of concentration in this case will depend on the flow from upstream, $Q_o$, and rates of lateral inflow per unit length, $q_l$. Integrating the inverse of velocity from an upstream discharge of $Q_o$ to a downstream discharge of $Q_o + Lq_l$ gives a time of concentration

$$TOC_v = \frac{L}{a} + \left[\frac{k}{q_l a}\left\{\ln\left(Q_o + Lq_l\right) - \ln Q_o\right\}\right] \qquad [A21]$$


Since $a$ is an asymptotic velocity at high discharges in this case, the time of concentration will approach the time to equilibrium at high flows.   This is illustrated in Figure A3.

Note that these determinations of both time of concentration and time to equilibrium are dependent on an assumption of a
continuous uniform rainfall.   They thus are representative only of rising limb velocities and celerities.   For full hydrograph prediction it will also be necessary to take account of the nonlinear nature of falling limb celerities.

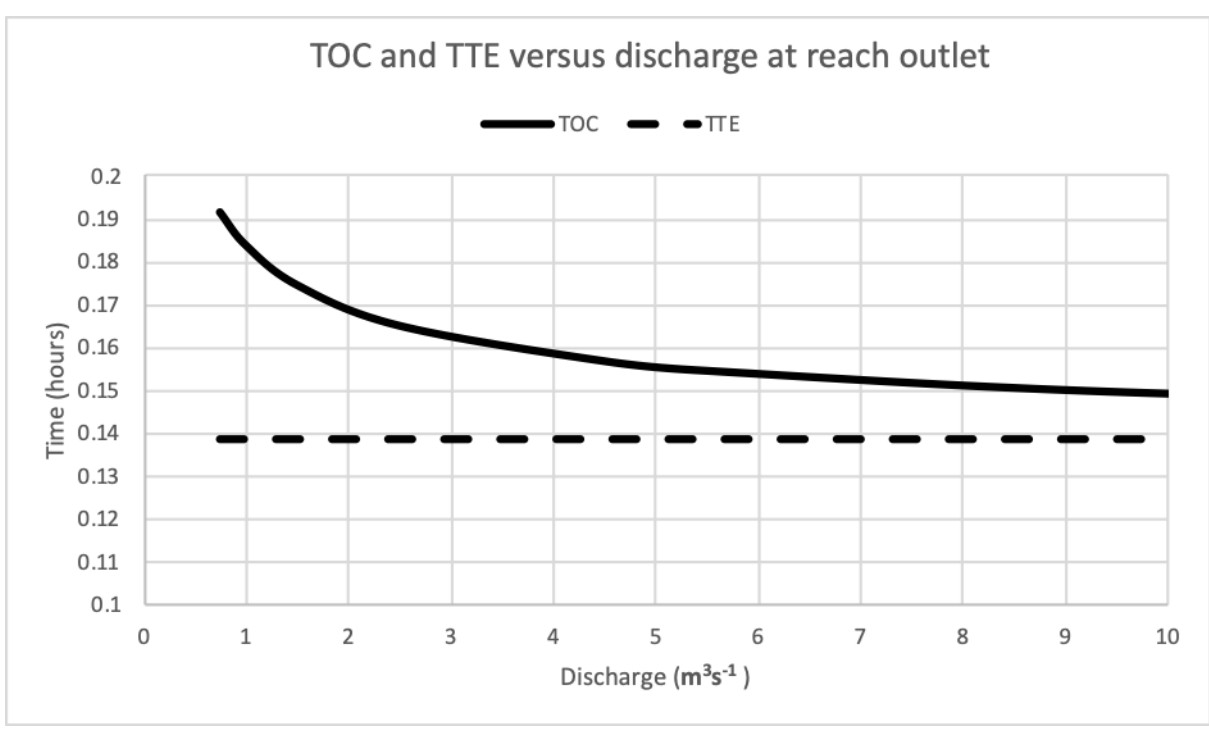





**Figure A3.  Time of Concentration (solid line) and Time to Equilibrium (dashed line) for an upland channel reach of 500m, with parameters from Beven (1979) (a = 1 m/s; k = 0.233 m³s⁻¹).   Upstream discharge 0.5 m³s⁻¹ ; lateral inflows from 0.0005 to 0.02 m²s⁻¹) (Equations [A20] and [A21])**

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
