# Peer review of "A history of the concept of time of concentration"

_Hydrology and Earth System Sciences, 2019_

## Referee Comment (RC1) · Anonymous Referee #1 · 21 Jan 2020

This paper seems to have two main purposes: first, it provides a historical overview of the concept of time of concentration. Second, it argues how this concept is misleading in its current definition. This seems to be a useful contribution, because (as the paper outlines), time of concentration has long been part of hydrology, but its use is often incorrect. The article is structured somewhat atypical, but the format works for me (though at times I was unsure where the paper was going). There is very little to disagree with in the paper and no new results are provided. Therefore, I think this paper can be published almost as is.

- The historical overview of past works is especially useful as past works are often forgotten now 1000s of hydrology papers are published annually. However, at times I was hoping that a more systematic overview (e.g. a table) of key papers was provided in addition to the story tying old papers together. However, I understand this may not be the purpose of a paper for this special issue (and more appropriate for a review

paper).

- Can it somewhere be made clear if the outlined misunderstanding of the time of concentration underlies the big quantitative differences (i.e. 500%) that have been reported by Grimaldi 2012?

- L25: would it be useful to have a proper Wikipedia citation that also includes the time and date this page was accessed, because the Wikipedia page may change over time (for example due to this article) https://en.wikipedia.org/wiki/Wikipedia:Citing_Wikipedia

Line 370: replace "Berghuis" by "Berghuijs"

---

## Referee Comment (RC2) · Anonymous Referee #2 · 28 Jan 2020

This is a very well written paper of interesting content as a review, not as a scientific paper. The appendix is an interesting comparison of calculation methods but does not add new knowledge.

However, I think it is addressing a problem that does not really exist and is mis-titled. It is really a history of rainfall to runoff hydrograph construction and manipulation, and the discussion of Time of Concentration seems somewhat bolted on to the main content. The fact of the matter is that hydrologists have been using simplifying mathematics to describe rather complicated natural phenomena in practical ways for a great many years, and on the whole these have proved very useful. Yet the author seems to have an "axe to grind" about this particular issue.

I recommend that it is re-written to sound less perjorative and re-titled as an historial survey of rainfall to runoff modelling, and simply note in the text that the term "Time of

[Figure]

Concentration" is badly used if taken too literally.

---

## Author Comment (AC1) · 6 Feb 2020

- The historical overview of past works is especially useful as past works are often forgotten now 1000s of hydrology papers are published annually. However, at times I was hoping that a more systematic overview (e.g. a table) of key papers was provided in addition to the story tying old papers together. However, I understand this may not be the purpose of a paper for this special issue (and more appropriate for a review

Response: Clearly there have been a huge number of papers that refer to time of concentration so a Table, even of key papers, would be rather long. A number of previous review papers are cited, including some relatively recent ones. These did not really discuss the different interpretations brought out here, or account for the resulting differences in estimates (as with the Grimaldi et al. paper noted below). So I think that telling the historical story and making that differentiation is the important aspect of this

paper.

- Can it somewhere be made clear if the outlined misunderstanding of the time of concentration underlies the big quantitative differences (i.e. 500%) that have been reported by Grimaldi 2012?

Response: Will do in revision

- L25: would it be useful to have a proper Wikipedia citation that also includes the time and date this page was accessed, because the Wikipedia page may change over time (for example due to this article) https://en.wikipedia.org/wiki/Wikipedia:Citing_Wikipedia

Response: Of course. Will give latest access when revising paper

Line 370: replace "Berghuis" by "Berghuijs"

Response: Thanks for catching that

k
* * *

---

## Author Comment (AC2) · 6 Feb 2020

This is a very well written paper of interesting content as a review, not as a scientific paper. The appendix is an interesting comparison of calculation methods but does not add new knowledge. However, I think it is addressing a problem that does not really exist and is mis-titled. It is really a history of rainfall to runoff hydrograph construction and manipulation, and the discussion of Time of Concentration seems somewhat bolted on to the main content. The fact of the matter is that hydrologists have been using simplifying mathematics to describe rather complicated natural phenomena in practical ways for a great many years, and on the whole these have proved very useful. Yet the author seems to have an "axe to grind" about this particular issue. I recommend that it is re-written to sound less perjorative and re-titled as an historial survey of rainfall to runoff modelling, and simply note in the text that the term "Time of Concentration" is badly used if taken too literally.

[Figure]

Response: Gven the material that is presented in this paper, this referee comment is really rather surprising. In particular:

1. This is not a history of rainfall-runoff models. A history of rainfall-runoff models would be much much longer (see Chapter 2 in my book on Rainfall-Runoff Modelling that gives a more extensive overview). Instead, as the title says, it concentrates specifically on the different and confused ways in which time of concentration has been used in the past which underpins the wide range of estimates that arise in its application.

2. In what way is the paper pejorative? It simply makes the distinction that definitions based on both velocities and celerities have been confused in the past, and suggests that we should be more careful in the use of the term time of concentration. If that is considered as an "axe to grind" then so be it - surely we should aim to apply concepts correctly!!

3. The Appendix is not a comparison of calculation methods, it provides derivations of time of concentrations under the kinematic wave assumptions for different surface and subsurface flow assumptions. Comparisons of calculation methods are given in the other review papers cited.

---

## Editor Comment (EC1) · Maurits Ertsen (Editor) · 25 Feb 2020

The length of the two reviews suggests that the paper does not have many issues. However, whereas that may be the case for one of them, there is the other review that poses a more principal question. Is this paper about time of concentration or on rainfall-runoff? In the background, this second review also poses a question that might concern all of us working on history of hydrology: the way hydrologists (and other water scholars) have used mathematics to make water work for them, even when the mathematics is not entirely "correct".

I do understand the response of the author to the question whether this is about rainfall-runoff. However, stating that it is not may not be enough in itself. I do not think that the paper needs to be rewritten, but a slightly more extensive engagement with the argumentation of the second reviewer would be welcome to position the paper in the

general discussion.

It is also clear that the paper is dealing with a historical phenomenon, but does so without taking a typical historical angle of studying how the societal issues of the time were related to the work of hydrologists, how professional struggles may have influenced the outcome of scholarly debates, etcetera. As such, the paper tends to isolate the debate on time of concentration somewhat from the larger historical contexts. However, the paper does show that something like a mathematical approach to catch an observation from the field has to be understood as a historical product with its own trajectory of debates, disagreements and changes.

---

## Author Comment (AC3) · 29 Feb 2020

I am not quite sure how to respond to this - even to the comment that "the paper is dealing with a historical phenomenon, but does so without taking a typical historical angle of studying how the societal issues of the time were related to the work of hydrologists, how professional struggles may have influenced the outcome of scholarly debates, etcetera" when I have shown how the origins of the time of concentration concept lay in the applied need to predict peak flows; how this was then developed into distributed approaches (with confusion between velocities and celerities); then simplified to unit hydrographic approaches to meet the same need (with continued confusion even in the context of the GUH); and that even recent reviews of the concept remain confused. As far as I can tell there has not been much debate about whether velocities or celerities should be used, only continuing confusion in how time of concentration is defined (with the wrong use of velocity-based definitions right from the beginning). While time

of concentration is a concept that underlies the estimation of peak discharges, the paper is definitely not about rainfall-runoff modelling. There are clearly other aspects to rainfall-runoff modelling that are not treated here - the whole paper is focused on time of concentration alone.

The editor does make an interesting comment about the use of approximate mathematics by hydrologists, but that is only partially relevant to the case about celerities and velocities being made here. These can be considered physical phenomena, with a long standing historical problem of comprehension by hydrologists, quite independent of the choice of mathematical assumptions used to describe them (again all the calculations in the appendix were only intended to provide illustrations of the differences that arise under different flow assumptions).

I can, of course, revise the paper to make these points more clearly. The comments also spurred me to try and find out more about the treatment in urban drainage in the late 19th Century to see if there was any use of celerities in defining concentration. This will take a little while longer to complete.
* * *

---

## Author Response (AR1)

Response to Editors Comments.

Dear Dr. Beven,

As you have seen, the reviewers are fairly positive on your paper. The main critical remarks came from reviewer 2. I think the discussion replies you provided may be a little too brief. Basically, you so "no, what rev 2 says is not the case", but your reasoning for that is limited. Would it be possible to include some further remarks about the debate you want to discuss in your paper versus two other debates relevant for history of hydrology, which are rainfall-runoff and the use of mathematical simplifications?

On the changes you already suggested before I made my decision (thanks for those), I have mixed feelings, to be honest. I have indicated before that a historical paper - that is a paper discussing why and how certain concepts were drafted and used - is not the place to solve an hydrological issue - that is an issue that needs a solution. Your new text does include quite some new judgements by yourself on the suitability of certain approaches. That suggests to me that the text keeps including two positions, a historical perspective on ideas and call for sound hydrology, that the historical discipline tries to keep separated. Would you be able to clarify your position on this problem of mixing?

Kind regards,

Maurits

I did previously reply to this issue in my first response to the Editor's comments (though I now see that this somehow ended up as a response to referee 2, though titled correctly.

I do not think my position has changed much.   The mix of historical perspective and call for sound hydrology seem to fit rather neatly together in this particular case.   In particular going back to the very first exposition of a time-area method for time of concentration, that due to Imbeaux at the end of the 19th Century, I have demonstrated that he used both velocities (for hillslope responses) and celerities (for routing in the channel network) in his methodology.   Once this is pointed out, therefore, it immediately requires some consideration of which should be used, especially as I show (historically) that the confusion persists to recent review papers without really being debated.   It is then only natural to make some recommendations about future practice in using the time of concentration concept.  I am not aware of any other paper in the literature that has considered this confused history of time of concentration.

I remain somewhat surprised at the reaction of Referee 2 who suggests that the paper is more a history of mathematical simplification in rainfall-runoff modelling, and seems rather blasé about what simplification is used.   The fact remains (as noted in the paper) that time of concentration is still frequently used in hydrological texts and papers which suggests that we should be clearer about how and why we use it (or, in the case of the WMO International Glossary Definition should not use it, despite its apparent common sense basis).   This was the original motivation for producing a history of the time of concentration (which, of course, has to carry over into its use in unit hydrograph theory, and the way in which DEMs have been used to produce times of concentration in this century).   I think this is also made clear in the abstract to the paper.

In producing this final revision, I have added some more material on early work on urban drainage systems and the issue of hydrograph skewness (after a communication from Walter Collischonn on the HESSD paper).

I would suggest that the paper is dominated by the historical development (Introduction section to Velocities, Celerities and Time of Concentration) and that this is suitably separated from the final relatively short sections on Mathematistry and Teaching).   I will accept that the Appendix is not essential to the historical part of the paper (that is why I moved it to an Appendix from the original submitted paper).   I would, however, like to keep it associated with this paper as potentially useful to anyone implementing the recommendations about teaching time of concentration / equilibrium concepts.